# Machine Learning Models for Point-of-Care Diagnostics of Acute Kidney Injury

**DOI:** 10.3390/diagnostics15212801

**Published:** 2025-11-05

**Authors:** Chun-You Chen, Te-I Chang, Cheng-Hsien Chen, Shih-Chang Hsu, Yen-Ling Chu, Nai-Jen Huang, Yuh-Mou Sue, Tso-Hsiao Chen, Feng-Yen Lin, Chun-Ming Shih, Po-Hsun Huang, Hui-Ling Hsieh, Chung-Te Liu

**Affiliations:** 1Graduate Institute of Biomedical Informatics, College of Medical Science and Technology, Taipei Medical University, Taipei 110, Taiwan; 99356@w.tmu.edu.tw; 2Artificial Intelligence Research and Development Center, Wan Fang Hospital, Taipei Medical University, Taipei 110, Taiwan; 3Department of Radiation Oncology, Wan Fang Hospital, Taipei Medical University, Taipei 116, Taiwan; 4Department of Surgery, School of Medicine, College of Medicine, Taipei Medical University, Taipei 110, Taiwan; 103164@w.tmu.edu.tw; 5Division of Cardiovascular Surgery, Department of Surgery, Wan Fang Hospital, Taipei Medical University, Taipei 116, Taiwan; 6Graduate Institute of Biomedical Electronics and Bioinformatics, National Taiwan University, Taipei 100, Taiwan; 7Department of Internal Medicine, School of Medicine, College of Medicine, Taipei Medical University, Taipei 110, Taiwan; hippy@tmu.edu.tw (C.-H.C.); sueym@tmu.edu.tw (Y.-M.S.); 88128@w.tmu.edu.tw (T.-H.C.); g870905@tmu.edu.tw (F.-Y.L.); cmshih53@gmail.com (C.-M.S.); 8Division of Nephrology, Department of Internal Medicine, Wan Fang Hospital, Taipei Medical University, Taipei 116, Taiwan; e516091013@tmu.edu.tw (Y.-L.C.); m118108006@tmu.edu.tw (N.-J.H.); 9Division of Nephrology, Department of Internal Medicine, Shuang Ho Hospital, Taipei Medical University, Taipei 235, Taiwan; 10Emergency Department, Department of Emergency and Critical Medicine, Wan Fang Hospital, Taipei Medical University, Taipei 116, Taiwan; 104228@w.tmu.edu.tw; 11Department of Emergency Medicine, School of Medicine, College of Medicine, Taipei Medical University, Taipei 110, Taiwan; 12Division of Cardiology and Cardiovascular Research Center, Department of Internal Medicine, Taipei Medical University Hospital, Taipei 110, Taiwan; 13Division of Cardiology, Department of Medicine, Taipei Veterans General Hospital, Taipei 112, Taiwan; huangbs@vghtpe.gov.tw; 14Second Degree Bachelor of Science in Nursing Collage of Medicine, National Taiwan University, Taipei 100, Taiwan; 15Department of Nursing, National Taiwan University Hospital Yunlin Branch, Yunlin 640, Taiwan

**Keywords:** acute kidney injury (AKI), artificial intelligence (AI), chronic kidney disease (CKD), creatinine, electronic alerts, intensive care units (ICU), hospitalization, machine learning

## Abstract

**Background/Objectives:** Computerized diagnostic algorithms could achieve early detection of acute kidney injury (AKI) only with available baseline serum creatinine (SCr). To tackle this weakness, we tried to construct a machine learning model for AKI diagnosis based on point-of-care clinical features regardless of baseline SCr. **Methods:** Patients with SCr > 1.3 mg/dL were recruited retrospectively from Wan Fang Hospital, Taipei. A Dataset A (*n* = 2846) was used as the training dataset and a Dataset B (*n* = 1331) was used as the testing dataset. Point-of-care features, including laboratory data and physical readings, were inputted into machine learning models. The repeated machine learning models randomly used 70% and 30% of Dataset A as training dataset and testing dataset for 1000 rounds, respectively. The single machine learning models used Dataset A as training dataset and Dataset B as testing dataset. A computerized algorithm for AKI diagnosis based on 1.5× increase in SCr and clinician’s AKI diagnosis compared to machine learning models. **Results:** On an independent, unbalanced test set (*n* = 1331), our machine learning models achieved AUROC values ranging from 0.67 to 0.74. A pre-existing computerized algorithm performed best (AUROC = 0.94). Crucially, all machine learning models significantly outperformed the routine clinician’s diagnosis (AUROC ~0.74 vs. 0.53, *p* < 0.05). For context, a pre-existing computerized algorithm, which requires available baseline SCr data, achieved an AUROC of 0.94 on a relevant subset of the data, highlighting the performance benchmark when baseline data is available. Formal statistical comparisons revealed that the top-performing models (e.g., Random Forest, SVM) were often statistically indistinguishable. Model performance was highly dependent on the test scenario, with precision and F1 scores improving markedly on a balanced dataset. **Conclusions:** In the absence of baseline SCr, machine learning models can diagnose AKI with significantly greater accuracy than routine clinical diagnoses. Our robust statistical analysis suggests that several advanced algorithms achieve a similarly high level of performance.

## 1. Introduction

Acute kidney injury (AKI) represents a significant adverse event among hospitalized patients [1]. Given that kidney function is frequently affected by cardiovascular dysfunction [2], sepsis [3], autoimmune diseases, and various causes of circulatory collapse [4], AKI serves as a critical indicator of in-hospital mortality and prolonged hospital stays [5]. Consequently, the early detection of AKI is a strategic approach for enhancing patient outcomes in hospital settings [6,7].

The implementation of an electronic AKI alert system holds considerable promise for early detection. A multicenter cohort study demonstrated that such a system improved renal function recovery in patients admitted to the intensive care units [8]. Another multicenter study reported that the use of a computerized decision support system was associated with reduced in-hospital mortality, fewer dialysis sessions, and shorter hospital stays [9]. In a prospective study conducted in Korea, an electronic AKI alert system linked to automated nephrologist consultation revealed that early consultation and intervention by a nephrologist increased the likelihood of renal recovery from AKI in hospitalized patients [10]. Although the electronic AKI alert system does not consistently alter clinical management or improve AKI outcomes, it shows potential for optimization [11,12].

Nevertheless, constructing an electronic AKI alert system is challenging. Such computerized algorithms detect AKI events based on an increase in serum creatinine (SCr) exceeding 1.5 times the baseline level within 7 days [13,14]. For patients lacking “baseline SCr within 7 days,” computerized algorithms must deduce a diagnosis, potentially lowering diagnostic accuracy. To address this limitation, machine learning models that detect AKI based on point-of-care clinical features may offer a solution. AKI is considered an ideal syndrome for the application of artificial intelligence due to its standardized and readily identifiable definition [15]. Patients with AKI may exhibit various clinical features, including demographic characteristics, comorbidities, changes in vital signs, and diverse laboratory findings [16]. These features may be input into machine learning models to facilitate point-of-care AKI diagnoses in the absence of baseline SCr levels. For instance, when a patient with no known medical history presents with abnormal SCr, it can be challenging for an inexperienced physician to differentiate between AKI and CKD, and to initiate an accurate diagnostic and therapeutic plan. In such cases, our machine learning model may assist less experienced clinicians in making this distinction, thereby enabling timely and appropriate clinical decision-making. Therefore, the present study aimed to construct machine learning models in the context of absent baseline SCr within seven days, utilizing clinical features at a single time point. To achieve this, we first assess model stability using a repeated sampling methodology (Method 1) before evaluating their real-world generalizability on an independent, temporally distinct test set (Method 2). We then benchmark these models against both routine clinician diagnoses and the traditional computerized algorithm to fully characterize their clinical potential.

## 2. Methods

### 2.1. Study Design and Participants

This retrospective study was conducted at Wan Fang Hospital, Taipei Medical University, Taipei, Taiwan. The study was approved by the Ethics Committee and Institutional Review Board of Taipei Medical University (approval no. N202111017, date 30 September 2021) and adhered to the tenets of the 1975 Declaration of Helsinki, as revised in 2013. Informed consent for participation is not required as per the Ethics Committee and Institutional Review Board of Taipei Medical University. The study population comprised hospitalized patients possessing one or more records of SCr levels exceeding 1.3 mg/dL. Two datasets, designated as Datasets A and B, were used for patients meeting this criterion. Dataset A served both as the training and testing set in the repeated machine learning model (Method 1), and as the training set for the single machine learning model (Method 2). Dataset B was exclusively used as the testing dataset for the single machine-learning model (Method 2). To mitigate selection bias, all patients were included through simple randomization. Given that 26 features were input into the machine learning models, the study aimed to enroll over 2600 patients in the training dataset and more than 1000 patients in the testing dataset.

The study was intentionally designed to use two separate datasets—Dataset A for training and Dataset B for testing—to ensure a rigorous and clinically relevant evaluation of the models. This approach serves two primary purposes. First, it facilitates temporal validation, where the models are trained on older data (Jan 2018–June 2020) and evaluated on newer, unseen data (July 2020–Dec 2020). This simulates a real-world deployment scenario and tests the model’s robustness to potential shifts in patient characteristics or clinical practices over time.

For Dataset A, patients hospitalized from January 2018 to June 2020 were randomly screened for eligibility for inclusion in the training dataset. The inclusion criteria were as follows: (1) at least one SCr value > 1.3 mg/dL during hospitalization and (2) age > 20 years. The exclusion criterion was the absence of baseline SCr within 7 days preceding the indexed abnormal SCr level. Patients with abnormal SCr levels were categorized into AKI and non-AKI groups using a computerized algorithm for AKI diagnosis, which will be detailed subsequently. The AKI and non-AKI groups were then randomly balanced to form Dataset A, with 1423 patients in each group.

For Dataset B, patients hospitalized from July 2020 to December 2020 with (1) at least one SCr value > 1.3 mg/dL during hospitalization and (2) age > 20 years were randomly selected for inclusion in the testing dataset. Notably, the availability of baseline SCr within 7 days was not a requirement for Dataset B. For each patient in Dataset B, a final diagnosis of AKI or non-AKI was retrospectively established by our researcher nephrologists based on a comprehensive review of the patient’s record according to KDIGO guidelines; this expert diagnosis served as the ground truth for evaluating all other methods. AKI was defined by the following criteria: (1) For patients with available baseline SCr values within 7 days before the indexed abnormal SCr level, an increase in SCr > 1.5 times satisfied the diagnosis of AKI. (2) For patients with available SCr values more than 7 days before the indexed abnormal SCr, the nearest previous SCr value was assumed to be the baseline SCr, and an increase in SCr > 1.5 times above this baseline value satisfied the diagnosis of AKI. 3. For cases in which previous SCr values were unavailable, it was assumed that patients had normal baseline SCr levels, and AKI was arbitrarily defined. Notably, the approach to patients without baseline SCr within 7 days stated above may lead to an AKI diagnosis in some CKD patients. In addition to this ground truth, two baseline diagnostic labels were collected for comparison: the routine clinician’s diagnosis documented in the final discharge summary, and the diagnosis generated by our pre-existing computerized algorithm. Dataset B included 334 and 997 patients in the AKI and non-AKI groups, respectively. To address the imbalance in Dataset B, additional trials using balanced and refined subsets were conducted. These trials showed improved or consistent model performance, indicating the imbalance was mitigated. Thus, the impact of dataset imbalance on model metrics was carefully evaluated and controlled.

### 2.2. Features of the Machine Learning Models

The features utilized in the machine learning models were sex, age, and laboratory and physical readings obtained at the time of admission. Notably, the present study aimed to diagnose AKI in patients with no known medical history who presented with abnormal SCr. Consequently, comorbidities were excluded from the feature set used by the machine learning model. Laboratory parameters included SCr, Na, K, aspartate aminotransferase (AST), alanine aminotransferase (ALT), red blood cell (RBC) count, hemoglobin, hematocrit (Hct), red cell distribution width (RDW-CV), white blood cell (WBC) count, and fractions of neutrophils, lymphocytes, monocytes, eosinophils, basophils, platelet counts, and platelet distribution width (PDW). Physical readings included respiration rate, systolic blood pressure (SBP), diastolic blood pressure (DBP), oxygen saturation (SpO_2_), body temperature, pulse rate, weight, and height. Before running the machine learning models, the association between the model features and AKI events was evaluated using principal component analysis (PCA), Uniform Manifold Approximation and Projection (UMAP). Physical readings were recorded by attending nurses at 7:00 on the day the indexed laboratory data were obtained. Biochemical data were measured using a Beckman Coulter DxC AU5800 (Beckman Coulter Inc., Brea, CA, USA), and hematological data were measured using a Beckman Coulter DxH 1601(Beckman Coulter Inc., Brea, CA, USA).

### 2.3. Computerized Algorithm for Defining AKI

The accuracy of the computerized algorithm was validated in a separate study conducted by our research team [15]. Briefly, if an azotemic patient (SCr > 1.3 mg/dL) with previous SCr value within 90 days was identified, an increase in SCr > 1.5 times satisfied the diagnosis of AKI; if an azotemic patient without previous SCr value within 90 days was identified, the baseline SCr was assumed to be normal and AKI was diagnosed arbitrarily; for an azotemic patient with previous SCr > 90 days before the indexed abnormal SCr, the nearest previous SCr was assumed as the baseline value, and increase in SCr > 1.5 times satisfied the AKI diagnosis. The program code was developed using Node.js 14.19.1 (OpenJS Foundation, San Francisco, CA, USA).

### 2.4. Researcher’s Definition for AKI and Clinician’s Diagnosis

AKI was defined according to the KDIGO Clinical Practice Guidelines for AKI [17]. Once an SCr value > 1.3 mg/dL was identified, the previous SCr values were reviewed. AKI was defined as follows: (1) In patients with previous SCr tests within 7 days preceding the indexed SCr values, an increase in SCr > 1.5 was defined as AKI. However, no cases of AKI were observed in the present study. (2) In patients with previous SCr tests exceeding 7 days before the indexed abnormal SCr values, the nearest previous SCr value was assumed as the baseline SCr, and an increase in SCr > 1.5 times above this value was defined as AKI. However, AKI was not observed in the present study. (3) In patients without previous SCr values, the patient was assumed to have normal baseline SCr levels, and AKI was defined as present. This diagnosis was utilized as the standard in the present study. Notably, the AKI criteria for decreased urine output in the KDIGO Clinical Practice Guidelines were not applied in this study. The AKI diagnosis documented in the discharge summaries was considered as the clinician’s diagnosis. In cases in which AKI was not included in the discharge diagnosis of patients with AKI, the diagnosis was considered inaccurate.

### 2.5. Data Preprocessing and Imputation

A standardized preprocessing pipeline was applied to the data before model training to ensure consistency and optimal performance. This pipeline, which includes imputation and feature scaling, was developed using only the training dataset (Dataset A) to prevent any information leakage from the test set (Dataset B).

First, to handle missing data, we employed mean imputation. In this procedure, any missing value for a given feature was replaced with the arithmetic mean of all observed values of that feature within the training data. This approach ensures a complete dataset for model training.

Second, following imputation, all features were standardized. This transformation rescales each feature so that it has a mean of zero and a standard deviation of one. Standardization is a critical step that prevents features with larger numeric ranges from disproportionately influencing the model’s learning process, which is particularly important for distance-based and gradient-based algorithms.

### 2.6. Development of Machine Learning Models

We employed two distinct approaches to develop and validate our machine learning models. *Method 1 (Repeated Learning)* was designed to assess the inherent stability and internal validity of the models by repeatedly training and testing on subsets of a single dataset (Dataset A). In contrast, *Method 2 (Single Learning on an Independent Test Set)* was designed to assess the models’ real-world generalizability on new, unseen data (Dataset B) and to investigate the impact of factors like data imbalance.

For Method 1, only Dataset A was used in the repeated machine learning models. In each iteration, 70% of Dataset A was randomly selected as the training dataset and the remaining 30% served as the testing dataset. This procedure was repeated 1000 times to obtain the average performance of the machine learning models. In each iteration, seven machine learning models were employed: the Support Vector Machine (SVM), Logistic Regression (LR), Gradient Boosting (GB), Extreme Gradient Boosting (XGBoost), Random Forest (RF), Naive Bayes classifier (NB), and Neural Network (NN).

For Method 2, we used the entire Dataset A as the training dataset to build a single, final version of each model. The performance of these models was then evaluated on the independent Dataset B, which served exclusively as the testing dataset.

The evaluation in Method 2 was conducted under three distinct clinical scenarios using the independent Dataset B:

Trial 1: Performance on the Full, Unbalanced Test Set

First, we tested the models on the entire Dataset B to evaluate their performance in a scenario that mirrors a typical clinical setting, where non-AKI cases are often more prevalent than AKI cases.

Trial 2: Performance on a Balanced Test Set

Second, to mitigate the potential effects of class imbalance on performance metrics like precision and F1-score, we evaluated the models on a reduced subset of Dataset B containing a balanced number of AKI and non-AKI patients.

Trial 3: Performance on a Post-Exclusion Test Set

Finally, to assess model performance on the most diagnostically definitive cases, we used a third subset of Dataset B that excluded patients who lacked baseline SCr values within the preceding seven days. This created a “cleaner” dataset to test the models’ core diagnostic capability.

The models were implemented using Python’sversion 3.12.12 scikit-learn and XGBoost libraries. For the more complex models, key hyperparameters were selected following a tuning process that utilized a randomized search with 5-fold cross-validation on the training data to optimize for accuracy. Specifically, the Random Forest model was constructed as an ensemble of 400 decision trees, with a maximum tree depth of 14 and a maximum of 8 features considered at each split. The XGBoost model was configured with 250 boosting rounds, a learning rate of 0.06, and a maximum tree depth of 4. Similarly, the Gradient Boosting model used 60 estimators and a learning rate of 0.11. The Neural Network was a Multi-layer Perceptron configured with a single hidden layer of 100 neurons and was set to stop training early if validation performance ceased to improve over 10 consecutive epochs to prevent overfitting. For the Support Vector Machine, Logistic Regression, and Naive Bayes models, we utilized the standard, well-established default parameters from the scikit-learn library, as they provided robust baseline performance without extensive modification.

### 2.7. Evaluation of Model Performance

The performance of each machine learning model was evaluated based on its accuracy, precision, recall (sensitivity), specificity, and F1 score calculated using the formula (2 × precision × recall)/(precision + recall). The predictive values of the machine learning models were evaluated using the area under the receiver operating characteristic curve (AUROC). These parameters constituted multiple performance metrics—accuracy, precision, recall, specificity, F1 score, and AUROC—to capture different aspects of diagnostic performance, especially under data imbalance. These metrics collectively assess the trade-offs between false positives and false negatives, which are critical in AKI diagnosis.

### 2.8. Statistical Analyses

Continuous variables with normal distribution were shown as mean ± standard deviation; continuous variables deviated from normal distribution were shown as median and interquartile range; categorical variables were shown as frequency and percentage. Analytic statistical tests for continuous variables with normal distribution were performed by using two-tailed *t*-test for independent samples; analytic statistical tests for continuous variables deviated from normal distribution were performed by using Wilcoxon sum rank test; analytic statistical tests for categorical variables were made using chi-squared test. *p* values of <0.05 was considered as significant. The distribution of data was examined using Q-Q plots. Statistical analysis was performed using SAS 9.4 (SAS Institute Inc., Cary, NC, USA).

To compare the performance between all machine learning models and baseline algorithms, we conducted pairwise statistical tests using the expert nephrologists’ diagnosis as the ground truth. For accuracy, recall (sensitivity), and specificity, we used McNemar’s test. For precision and F1-score, we employed a non-parametric bootstrap procedure with 2000 resamples. For the area under the receiver operating characteristic curve (AUROC), we used DeLong’s test. A *p*-value of <0.05 was considered statistically significant. The results of these pairwise comparisons are presented in the results tables using superscript letter notations, where models sharing a letter are not significantly different from one another.

## 3. Results

### 3.1. Clinical Characteristics of the Datasets

Dataset A comprised 1423 patients with AKI and 1423 without AKI. Sex, point-of-care SCr, RBC, Hct, body temperature, weight, and height were similar between AKI and non-AKI groups. In contrast, AKI patients were significantly younger; had higher serum Na and K; higher AST and ALT; lower hemoglobin; higher RDW-CV; higher WBC count; higher neutrophil fraction; lower lymphocyte, monocyte, eosinophil, and basophil fractions; lower platelet counts; and higher PDW. Patients with AKI had significantly higher respiratory rates, lower SBP and DBP, lower SpO_2_ and higher pulse rates (Table 1).

Dataset B consisted of 1331 hospitalized patients with abnormal renal function (SCr > 1.3 mg/dL), categorized into 334 patients with AKI and 997 non-AKI patients by the nephrologists involved in this study. The AKI and non-AKI groups exhibited similar sex, age, serum K, RDW-CV, body temperature, and height. Patients with AKI had significantly lower SCr, higher levels of serum Na, AST and ALT, RBC, hemoglobin, and Hct; higher WBC, neutrophil fraction, lower lymphocyte, monocyte, eosinophil, and basophil fractions; lower platelet count and higher PDW; higher respiratory rate; lower SBP and DBP; lower SpO_2_; and higher pulse rate (Table 2).

PCA and UMAP were used to reduce the dimensions of all the features mentioned above to visualize their correlations with AKI. Both PCA and UMAP showed that these features roughly distributed patients with AKI to the right upper dimension and non-AKI patients to the left lower dimension (Figure 1). The substantial overlap between the orange and blue clusters in both plots visually demonstrates the inherent difficulty in diagnosing AKI using point-of-care data. There is no simple, clear boundary that separates the two patient populations. This complexity underscores the limitations of simple diagnostic rules and provides the rationale for using machine learning models, which are designed to identify subtle, non-linear patterns within such complex data. The rough distribution, where AKI patients (orange) show some tendency to cluster, suggests that such patterns exist for a model to learn.

### 3.2. Repeated Machine Learning Model (Method 1)

As previously described, for Method 1, 70% of Dataset A was randomly selected for use as the training dataset and the remaining 30% was used as the testing dataset. This method was applied to all seven machine-learning models previously mentioned. This procedure was repeated 1000 times to obtain the average performance of the machine learning models. The performance metrics of each machine learning model are expressed as the mean ± standard deviation of 1000 iterations. The accuracy of Method 1 ranged from 0.65 0.69. The machine learning models SVM, GB, XGB, and RF achieved the highest accuracy (0.69 ± 0.01). The F1 score for Method 1 ranged from 0.55 0.69. The machine learning models XGB and RF achieved the highest F1 score (F1 score = 0.69 ± 0.01). The machine learning models of Method 1 exhibited AUROC ranging from 0.73 to 0.76, of which SVM, GB, XGB, and RF demonstrated the highest AUROC (0.76 ± 0.01). Overall, using Method 1, the machine models XGB and RF exhibited the best performance (Table 3).

### 3.3. Single Machine Learning Models on an Independent Test Set (Method 2)

We evaluated the seven trained machine learning models and two baseline methods on the independent test set (Dataset B), using the researcher nephrologists’ diagnosis as the ground truth. The evaluation was conducted under three different clinical scenarios, with detailed performance metrics and statistical comparisons presented in Table 4, Table 5 and Table 6.

Performance on the Unbalanced Test Set: In the first trial, using the entire, unbalanced Dataset B (*n* = 1331), the models achieved AUROC values from 0.67 to 0.74 (Table 4). The computerized algorithm was the top performer (AUROC 0.94). All ML models demonstrated statistically significantly superior performance compared to the clinician’s diagnosis across key metrics, for instance, F1 score (~0.52 vs. 0.35) and AUROC (~0.74 vs. 0.53). The letter notations indicate that the top-performing models, such as RF and SVM, were statistically indistinguishable from each other.

Performance on the Balanced Test Set: In the second trial, on a balanced subset (*n* = 334 in each group), there was a substantial improvement in precision and F1 scores (e.g., RF’s F1 score increased from 0.52 to 0.63) for most models (Table 5). This highlights the models’ learning capability when the effect of class imbalance is mitigated. The ML models remained significantly superior to the clinician’s diagnosis, again showing a clear advantage in both F1 score (e.g., ~0.63 vs. 0.50) and AUROC.

Performance on the Post-Exclusion Test Set: In the third trial, on a subset of the most definitive diagnoses (*n* = 398), the ML models’ AUROCs ranged from 0.64 to 0.71 (Table 6). Even in this “cleaner” data scenario, the ML models maintained a significant performance advantage over the clinician’s diagnosis in both F1 score (e.g., RF 0.59 vs. 0.46) and AUROC (e.g., RF 0.71 vs. 0.53).

### 3.4. Feature Importance of the Machine Learning Models

The three machine learning models that demonstrated the best performance in the third trial of Method 2, RF, XGBoost, and GB, were selected to evaluate the feature importance. In the RF model, the three features with the highest importance were lower lymphocyte fraction, SBP, and SCr (Figure 2A). In the XGBoost model, the three features with the highest importance were lower SCr level, lower lymphocyte fraction, and higher WBC count (Figure 2B). In the GB model, the three features with the highest importance were low SCr, high AST, and low SBP (Figure 2C). Overall, lower SCr, higher WBC count, lower lymphocyte fraction, higher RDW-CV, lower platelet count, higher GOT, younger age, lower SBP, higher pulse rate, and higher respiratory rate were features of AKI.

## 4. Discussion

In summary, the repeated machine learning models employed in the present study demonstrated accuracy ranging from 0.65 to 0.69 and an AUROC ranging from 0.73 to 0.76 for the diagnosis of AKI in the absence of baseline SCr. Conversely, the single machine leaning models exhibited an accuracy range of 0.53 to 0.74 and the AUROC ranged from 0.70 to 0.74 for the diagnosis of AKI without available baseline SCr. These findings suggest that repeated machine-learning models offer superior accuracy and predictive value for AKI diagnosis. Additionally, while the single machine learning models did not exhibit better accuracy and predictive value in the balanced testing dataset (Method 2, trial 2), they exhibited better performance in the testing dataset after exclusion of patients with uncertain AKI status (Methods 2, trial 3). Notably, with available past SCr records, the computerized algorithm exhibited superiority in every index compared to either repeated or single machine learning models. While RF, XGBoost, and GB consistently ranked among the top models, our statistical analysis revealed no significant performance difference between them across most metrics. This suggests that several advanced algorithms can achieve a similar performance ceiling. The results imply that future improvements may lie more in feature engineering and addressing data-driven challenges like class imbalance, rather than in selecting a single ‘best’ algorithm.

As the diagnosis of AKI is based on an increase in SCr over a 7-day period [16], computerized algorithms can accurately diagnose AKI in patients with available baseline SCr or a recent record of SCr. Nevertheless, for patients without such reference SCr values, the diagnosis of AKI is challenging using computerized algorithms, and even for clinicians. The present study attempted to overcome this impediment using machine learning models to identify AKI events based on point-of-care features of patients presenting with abnormal SCr. In cases where a patient with no known medical history presents with abnormal SCr, our machine learning model can assist inexperienced physicians in distinguishing between AKI and CKD, thereby facilitating the initiation of appropriate diagnostic and therapeutic strategies. Remarkably, all included patients had abnormal SCr values; thus, the function of our models was to distinguish AKI events from preexisting chronic kidney disease. To date, the application of machine learning in the management of AKI has primarily focused on AKI prediction. Thus, AKI prediction models with short time windows can be compared to our AKI diagnosis models [18]. In an AKI prediction model for all-care settings conducted by Cronin et al. in 2015 [19], pre-admission laboratory tests of −5 days to +48 h from the admission date were obtained for over 1.6 million hospitalizations for model training. They found that the models (LR, LASSO regression, and RF) exhibited an AUROC of 0.746–0.758 for predicting in-hospital AKI events [19]. In another study, He et al. tested machine learning models to differentiate AKI in different prediction time windows. Their models exhibited AUROC values ranging from 0.720 to 0.764. Among the tested models, the best model performance was achieved in predicting AKI one day in advance [20]. A similar study by Cheng et al. tested different data collection time windows to train datasets of AKI prediction models. The results suggested that the RF algorithm showed the best performance for AKI prediction 1–3 days in advance, with AUROC of 0.765, 0.733, and 0.709, respectively [21]. Compared with these studies, our repeated machine learning models exhibited AUROC of 0.73–0.76, depending on the training model used, showing that repeated machine learning could exhibit similar performance with different model algorithms and is comparable to machine learning models with large training datasets.

Although we compared the present AKI diagnostic model with AKI prediction models, a difference existed between these two models. Koyner et al. tested models with and without a change in SCr from the baseline in an all-care setting. The results showed that excluding “change in SCr” from input features did not affect the model’s AKI prediction ability [22]. In contrast, in the present study, SCr was an important feature to be input into the machine learning model for AKI diagnosis, regardless of the algorithm used. The reason for this difference may be that AKI prediction relies more on the severity of comorbidities than on the existing abnormal SCr readings. In contrast, the SCr value at point-of-care is an important feature for the identification of AKI events.

Among the studies developing AKI prediction models, researchers have been seeking the best machine learning algorithm for predicting the risk of AKI events. The AKI prediction model developed by Cronin et al. in 2015, using a 1.6 million training dataset revealed that the performance of traditional LR and LASSO regression models was slightly superior to that of the RF model [19]. In a 2021 study by Kim et al., in which they intended to develop a continuous real-time prediction model for AKI events, a recurrent neural network algorithm was found to be most suitable for predicting AKI events 48 h in advance [23]. In the single machine learning models used in the present study, we found that the RF, XGBoost, and GB algorithms exhibited superior performance in AKI diagnosis. Nonetheless, in the case of repeated machine learning models, the differences between the different algorithms were not evident. This finding suggests that with repeated training, the performance of different machine-learning algorithms may approach a consistent level.

Yue et al. developed a machine learning model for AKI prediction in patients with sepsis, identifying key features such as urine output, mechanical ventilation, body mass index, estimated glomerular filtration rate, SCr, partial thromboplastin time, and blood urea nitrogen [24]. In addition to features directly associated with renal function, those indicative of general disease severity are crucial in this model of sepsis-related AKI. In the present model, which was designed for an all-care setting, features related to sepsis include lymphocyte fraction, white blood cell (WBC) count, platelet count, pulse rate, SBP, and GOT also play important roles. This finding suggests that, in an all-care setting, sepsis is the most important cause of AKI in hospitalized patients.

As electronic diagnostic tools have been integrated into decision support and electronic alert systems for AKI, these studies showed a heterogeneous system design and revealed mixed results [25]. Previous research has shown that electronic AKI alert systems possess acceptable accuracy and applicability [26,27]. Furthermore, Hodgson et al. demonstrated that their electronic AKI alert system reduced the incidence of hospital-acquired AKI and in-hospital mortality [28]. Conversely, a study by Wilson et al. involving 6030 patients indicated that the electronic AKI alert system did not reduce the risk of the primary outcome, with variable effects across clinical centers [29]. The findings of the present study suggest that, while electronic diagnostic tools may enhance the accuracy of AKI diagnosis, timely differential diagnosis and management are imperative to improve outcomes.

The results of the present study showed that single model trials of machine learning models were associated with a wide variety of accuracy. The variation in accuracy (0.53–0.74) across single model trials reflects differences in Dataset B’s characteristics—specifically, data imbalance and inclusion of deduced diagnoses. When tested with balanced or refined subsets, model performance became more consistent (accuracy 0.63–0.72), with RF, XGBoost, and GB generally outperforming others. This demonstrates that the observed inconsistency is largely driven by dataset quality and composition. In addition, we also found that machine learning models underperform compared to traditional computerized algorithms in diagnosing AKI. A possible explanation may be that the computerized algorithm achieved higher accuracy (up to 0.95) because it directly relied on detecting a defined increase in baseline serum creatinine (SCr), as per AKI diagnostic criteria. In contrast, our machine learning models were designed for cases lacking recent baseline SCr, a scenario where traditional algorithms fail or rely on assumptions. A key finding of our study, now robustly supported by formal statistical analysis, is that machine learning models significantly outperform routine clinician’s diagnoses in identifying AKI when baseline SCr is unavailable. This was consistently observed across all three distinct evaluation scenarios (Table 4, Table 5 and Table 6). This suggests that in data-limited, real-world settings where clinicians may rely on subjective judgment, the models provide a valuable and more accurate diagnostic support tool.

The present study unexpectedly demonstrates that machine learning models outperform clinicians in diagnosing acute kidney injury (AKI). One possible explanation is that AKI may not have been the primary clinical concern in many cases, with clinicians focusing more on dominant conditions such as sepsis or heart failure. As a result, timely recognition and management of AKI were sometimes overlooked. Additionally, when patients presented with renal impairment but lacked baseline renal function data, clinicians often relied on subjective judgment to diagnose AKI. In such scenarios—where objective diagnostic criteria are unavailable—machine learning models offer valuable support, enabling timely and accurate decision-making. Moving forward, we aim to incorporate clinician feedback into model development to explore the root causes of misdiagnoses and further enhance diagnostic performance.

A limitation of the present study is the relatively small sample size, particularly for the testing dataset. However, considering the all-care setting in the present study, our machine learning models may be applied to hospitalized patients admitted to both critical care units and general wards. Another limitation was the single-center design that limits the generalizability of the results. To compensate for this, an independent testing dataset was used for validation (Dataset B). Nevertheless, external validation is restricted by the institutional review board and is therefore not feasible. Furthermore, this study did not exclude patients based on specific comorbidities. Clinical features used in our models, such as inflammatory markers and vital signs, can be influenced by a wide range of conditions beyond AKI, such as sepsis, heart failure, or diabetes. This could introduce confounding factors and affect model stability. However, this approach was intentional, as our goal was to develop models that could function in a real-world clinical setting where patients often present with multiple, complex health issues. The models were thus trained to identify diagnostic patterns within this inherent clinical complexity. Nevertheless, future research should aim to quantify the impact of specific comorbidities on model performance. Integrating established comorbidity indices, such as the Charlson Comorbidity Index, as input features could potentially improve model robustness and accuracy.

Looking ahead, several avenues for future research could build upon our findings. Future studies should explore more advanced machine learning algorithms, such as deep learning models or sophisticated ensembles, to potentially capture more complex relationships in the data. To improve the generalizability of these models, conducting multicenter studies with diverse data sources—including time-series clinical data and novel inputs like genetic markers—is crucial. Most importantly, prospective validation in real-time clinical settings is essential to assess the models’ true clinical impact, their utility in decision support, and their seamless integration into existing hospital workflows.

In conclusion, machine learning models were able to diagnose AKI without baseline SCr records. Additionally, the machine learning models for AKI diagnosis demonstrated superior accuracy compared to clinicians. Also, repeated machine learning models exhibited more consistent and superior performance than single machine learning models. Notably, the computerized AKI diagnostic algorithms showed superior accuracy compared to the machine learning models when baseline SCr was available. Therefore, these two approaches can be combined to develop a more comprehensive electronic AKI diagnostic system.

## Figures and Tables

**Figure 1 diagnostics-15-02801-f001:**
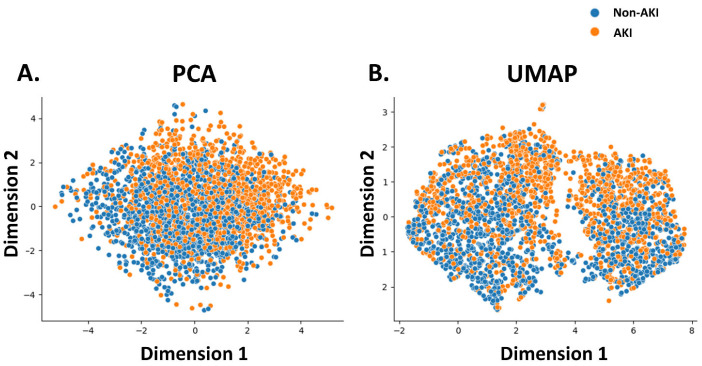
Feature visualization and dimensionality reduction. This figure aims to visualize the overall clinical data to see if patients with Acute Kidney Injury (AKI) and those without (Non-AKI) form distinct, separable groups. Since it is impossible to plot all 26 clinical features simultaneously, we use two dimensionality reduction techniques—(**A**) Principal Component Analysis (PCA) and (**B**) Uniform Manifold Approximation and Projection (UMAP)—to compress the complex, high-dimensional data into a simple two-dimensional map. Each dot on the plots represents a single patient, colored according to their final diagnosis: orange for AKI and blue for Non-AKI. This visualization highlights the substantial overlap between the AKI and non-AKI patient clusters, visually confirming the difficulty of simple linear separation and reinforcing the need for sophisticated machine learning models to diagnose AKI from point-of-care data. 2D: two-dimensional; PCA, Principal Component Analysis; UMAP, Uniform Manifold Approximation and Projection.

**Figure 2 diagnostics-15-02801-f002:**
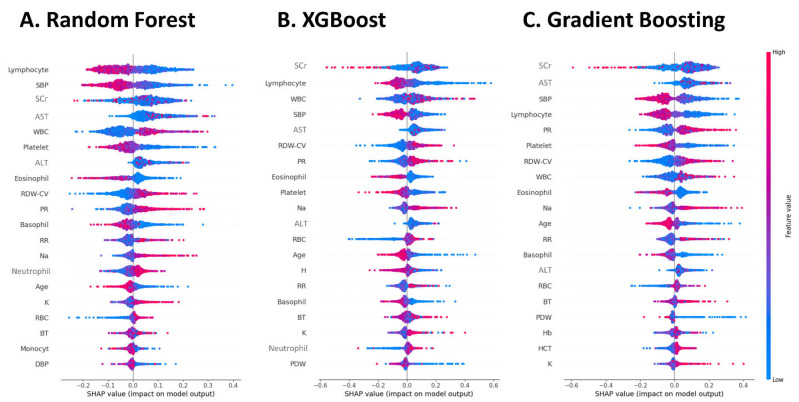
Feature importance of the three best performing machine learning models. This figure displays the most important clinical features for the three top-performing machine learning models, as determined by SHAP values. (**A**) Random Forest Model. (**B**) XGBoost Model. (**C**) Gradient Boosting Model. Feature Ranking (*Y*-axis): Clinical features are ranked from top to bottom based on their overall impact on the model’s predictions. Higher-ranked features are more influential. Impact on Diagnosis (*X*-axis): Each dot represents a single patient. A dot’s position on the horizontal axis shows how that feature value influenced the model’s conclusion for that patient. Dots on the right side (positive SHAP values) pushed the prediction towards an AKI diagnosis, while dots on the left side (negative SHAP values) pushed the prediction towards a non-AKI diagnosis. Feature Value (Color): The color of each dot indicates the feature’s value for that patient. Red dots represent higher clinical values (e.g., high serum creatinine), and blue dots represent lower values. SHAP value, SHapley Additive exPlanations; XGBoost, Extreme Gradient Boosting.

**Table 1 diagnostics-15-02801-t001:** Baseline characteristics of Dataset A.

Characteristic	AKI	Non-AKI	*p* Value
Number, *n*	1423	1423	n/a
Male, *n* (%)	822 (57)	854 (60)	0.13
Age, years	73.2 ± 16.1	75.1 ± 14.1	<0.05
Creatinine, mg/dL	3.6 ± 2.3	3.7 ± 3.2	0.30
Na, mmol/L	140.4 ± 9.1	138.5 ± 7.2	<0.05
K, mmol/L	4.4 ± 1.0	4.1 ± 0.7	<0.05
AST, U/L	33.0 (58.0)	23.0 (19.5)	<0.05
ALT, U/L	22.0 (33.5)	15 (18.0)	<0.05
RBC, 10^6^/µL	3.3 ± 0.8	3.4 ± 0.8	0.12
Hb, g/dL	10.0 ± 2.2	10.3 ± 2.1	<0.05
Hct, %	30.3 ± 6.9	30.8 ± 6.6	0.06
RDW-CV, %	16.8 ± 3.0	15.7 ± 2.4	<0.05
WBC, 10^3^/µL	13.1 ± 19.0	10.2 ± 13.5	<0.05
Neutrophil, %	80.0 ± 15.4	74.2 ± 14.3	<0.05
Lymphocyte, %	10.4 ± 11.3	14.5 ± 10.3	<0.05
Monocyte, %	6.2 ± 4.05	7.7 ± 4.1	<0.05
Eosinophil, %	1.1 ± 2.1	2.2 ± 3.2	<0.05
Basophil, %	0.3 ± 0.5	0.5 ± 0.5	<0.05
Platelet, 10^6^/µL	135 (136)	181 (120)	<0.05
PDW, %	17.4 ± 0.8	17.2 ± 0.7	<0.05
Respiratory rate, /min	19.5 ± 5.3	18.3 ± 3.5	<0.05
SBP, mmHg	117.2 ± 23.9	127.9 ± 22.4	<0.05
DBP, mmHg	64.7 ± 16.2	68.5 ± 14.1	<0.05
SpO_2_, %	96.4 ± 4.4	96.9 ± 3.1	<0.05
Temperature, °C	36.7 ± 0.6	36.7 ± 0.4	0.25
Pulse rate, /min	91.8 ± 21.4	83.4 ± 17.5	<0.05
Weight, kg	61.2 ± 15.2	62.2 ± 14.8	0.09
Height, cm	158.6 ± 15.3	159.1 ± 10.3	0.32

Dataset A comprised randomly selected AKI and non-AKI hospitalized patients (*n* = 1423 in each group). The presence or absence of AKI was determined using a validated computerized algorithm. AKI, acute kidney injury; AST, aspartate aminotransferase; ALT, alanine aminotransferase; RBC, red blood cell; Hb, hemoglobin; Hct, hematocrit; RDW-CV, red cell distribution width; WBC, white blood cell; PDW, platelet distribution width; SBP, systolic blood pressure; DBP, diastolic blood pressure; SpO_2_, oxygen saturation. Categorical variables were expressed as frequency (%) and analyzed using the chi-square test; continuous variables with normal distribution were expressed as mean ± standard deviation and analyzed using Student’s *t* test for independent groups; continuous variables deviated from normal distribution were expressed as median (interquartile range) and analyzed using Wilcoxon sum rank test.

**Table 2 diagnostics-15-02801-t002:** Baseline characteristics of Dataset B.

Characteristics	AKI	Non-AKI	*p*-Value
Number, *n*	334	997	n/a
Male, *n* (%)	207 (62)	648 (65)	0.26
Age, years	73.3 ± 16.5	72.6 ± 14.3	0.45
Creatinine, mg/dL	3.0 ± 1.9	3.4 ± 2.9	<0.05
Na, mmol/L	139.2 ± 8.2	137.8 ± 6.4	<0.05
K, mmol/L	4.2 ± 0.8	4.2 ± 0.6	0.56
AST, U/L	26.5 (33.8)	19 (12)	<0.05
ALT, U/L	21 (30.6)	15 (13)	<0.05
RBC, 10^6^/µL	3.7 ± 0.8	3.6 ± 0.8	<0.05
Hb, g/dL	11.2 ± 2.5	10.8 ± 2.3	<0.05
Hct, %	33.5 ± 7.7	32.4 ± 7.1	<0.05
RDW-CV, %	15.6 ± 2.7	15.5 ± 2.4	0.81
WBC, 10^3^/µL	12.0 ± 7.5	8.17 ± 4.4	<0.05
Neutrophil, %	79.0 ± 13.5	70.6 ± 13.5	<0.05
Lymphocyte, %	11.5 ± 10.0	17.6 ± 10.3	<0.05
Monocyte, %	6.56 ± 4.3	8.44 ± 4.0	<0.05
Eosinophil, %	1.3 ± 2.5	2.4 ± 2.7	<0.05
Basophil, %	0.3 ± 0.4	0.6 ± 0.5	<0.05
Platelet, 10^6^/µL	171 (119)	194 (108)	<0.05
PDW, %	17.2 ± 0.8	17.1 ± 0.6	<0.05
Respiratory rate, /min	18.8 ± 6.5	17.7 ± 2.6	<0.05
SBP, mmHg	121.8 ± 24.4	130.8 ± 21.4	<0.05
DBP, mmHg	68.6 ± 16.2	71.8 ± 14.3	<0.05
SpO_2_, %	96.4 ± 4.5	97.2 ± 2.9	<0.05
Temperature, °C	36.7 ± 0.5	36.6 ± 0.3	0.11
Pulse rate, /min	86.4 ± 18.9	79.3 ± 14.7	<0.05
Weight, kg	64.9 ± 17.7	64.2 ± 14.7	0.48
Height, cm	161.4 ± 8.9	161.6 ± 9.0	0.67

Dataset B comprised 334 patients with AKI and 997 patients without AKI. AKI was diagnosed by nephrologists involved in the present study. AKI, acute kidney injury; AST, aspartate aminotransferase; ALT, alanine aminotransferase; RBC, red blood cell; Hb, hemoglobin; Hct, hematocrit; RDW-CV, red cell distribution width; WBC, white blood cell; PDW, platelet distribution width; SBP, systolic blood pressure; DBP, diastolic blood pressure; SpO_2_, oxygen saturation. Categorical variables were expressed as frequency (%) and analyzed using the chi-square test; continuous variables with normal distribution were expressed as mean ± standard deviation and analyzed using Student’s *t* test for independent groups; continuous variables deviated from normal distribution were expressed as median (interquartile range) and analyzed using Wilcoxon sum rank test.

**Table 3 diagnostics-15-02801-t003:** Repeated machine learning models based on Dataset A (*n* = 1423 in each group).

Model	Accuracy	Precision	Recall	Specificity	F1 Score	AUROC
SVM	0.69 ± 0.01	0.70 ± 0.01	0.67 ± 0.02	0.71 ± 0.02	0.68 ± 0.01	0.76 ± 0.01
LR	0.67 ± 0.01	0.68 ± 0.01	0.64 ± 0.02	0.70 ± 0.02	0.66 ± 0.01	0.73 ± 0.01
GB	0.69 ± 0.01	0.70 ± 0.01	0.67 ± 0.02	0.70 ± 0.02	0.68 ± 0.01	0.76 ± 0.01
XGB	0.69 ± 0.01	0.70 ± 0.01	0.68 ± 0.02	0.70 ± 0.02	0.69 ± 0.01	0.76 ± 0.01
RF	0.69 ± 0.01	0.69 ± 0.01	0.68 ± 0.02	0.70 ± 0.02	0.69 ± 0.01	0.76 ± 0.01
NB	0.65 ± 0.01	0.76 ± 0.02	0.44 ± 0.05	0.86 ± 0.02	0.55 ± 0.04	0.73 ± 0.01
NN	0.67 ± 0.01	0.68 ± 0.02	0.65 ± 0.02	0.70 ± 0.03	0.67 ± 0.01	0.74 ± 0.01

The parameters were derived from machine learning repeated 1000 times and expressed as the mean ± standard deviation. In each machine learning iteration, 70% of the data were randomly selected as the training dataset, and the remaining 30% were used as the testing dataset. The presence or absence of AKI was determined using a validated computerized algorithm. SVM, Support Vector Machine; LR, Logistic Regression; GB, Gradient Boosting; XGB, Extreme Gradient Boosting; RF, Random Forest; NB, Naive Bayes classifier; NN, Neural Network.

**Table 4 diagnostics-15-02801-t004:** Single machine learning models based on Dataset A (*n* = 1423 in each group) and unbalanced Dataset B as the testing dataset (*n* = 1331).

Model	Accuracy	Precision	Recall	Specificity	F1 Score	AUROC
SVM	0.74 ^a^	0.48 ^a^	0.52 ^abc^	0.81 ^ab^	0.50 ^a^	0.74 ^a^
RF	0.75 ^ab^	0.50 ^a^	0.54 ^ab^	0.82 ^ab^	0.52 ^a^	0.74 ^a^
GB	0.76 ^ab^	0.51 ^a^	0.54 ^ab^	0.83 ^b^	0.52 ^a^	0.73 ^ac^
NB	0.77 ^b^	0.58 ^b^	0.27 ^d^	0.93 ^c^	0.37 ^b^	0.73 ^abc^
XGB	0.75 ^ab^	0.49 ^a^	0.55 ^b^	0.81 ^a^	0.52 ^a^	0.72 ^b^
LR	0.74 ^a^	0.48 ^a^	0.49 ^ac^	0.82 ^ab^	0.49 ^a^	0.71 ^bc^
NN	0.75 ^ab^	0.51 ^ab^	0.23 ^d^	0.92 ^c^	0.32 ^b^	0.67 ^e^
Traditional methods						
Computerized algorithm	0.97 ^c^	0.98 ^c^	0.88 ^e^	0.99 ^e^	0.93 ^c^	n/a
Clinician’s diagnosis	0.57 ^d^	0.28 ^d^	0.46 ^c^	0.61 ^d^	0.35 ^b^	n/a

Performance of models on the full, unbalanced test set (Dataset B, *n* = 1331). Performance was evaluated using the researcher nephrologists’ diagnosis as the ground truth. Values in the same column sharing a common superscript letter are not statistically significantly different (*p* ≥ 0.05). Pairwise comparisons were performed using McNemar’s test (Accuracy, Recall, Specificity), DeLong’s test (AUROC), and a bootstrap procedure (Precision, F1-score). Within the same column, values followed by the same superscript letter (e.g., a, b, ab) are not statistically significantly different (*p* ≥ 0.05). Notably, this table demonstrates that even in an unbalanced, real-world scenario, all developed machine learning models delivered significantly better diagnostic performance (based on F1 score and AUROC) than the routine clinician’s diagnosis. SVM, Support Vector Machine; LR, Logistic Regression; GB, Gradient Boosting; XGB, Extreme Gradient Boosting; RF, Random Forest; NB, Naive Bayes classifier; NN, Neural Network.

**Table 5 diagnostics-15-02801-t005:** Single machine learning models based on Dataset A (*n* = 1423 in each group) and balanced Dataset B as the testing dataset (*n* = 334 in each group).

Model	Accuracy	Precision	Recall	Specificity	F1 Score	AUROC
GB	0.69 ^a^	0.77 ^a^	0.54 ^ab^	0.84 ^a^	0.63 ^a^	0.75 ^a^
RF	0.68 ^a^	0.76 ^a^	0.54 ^ab^	0.83 ^a^	0.63 ^a^	0.75 ^a^
SVM	0.67 ^a^	0.74 ^a^	0.52 ^abc^	0.81 ^a^	0.61 ^a^	0.75 ^a^
XGB	0.68 ^a^	0.75 ^a^	0.55 ^b^	0.81 ^a^	0.63 ^a^	0.73 ^a^
LR	0.66 ^a^	0.75 ^a^	0.49 ^ac^	0.84 ^a^	0.59 ^a^	0.73 ^a^
NB	0.61 ^b^	0.85 ^b^	0.27 ^d^	0.95 ^b^	0.41 ^b^	0.74^a^
NN	0.59 ^bc^	0.79 ^ab^	0.23 ^d^	0.94 ^b^	0.36 ^b^	0.68 ^c^
Traditional methods						
Computerized algorithm	0.94 ^d^	1.00 ^c^	0.88 ^e^	1.00 ^d^	0.93 ^d^	n/a
Clinician’s diagnosis	0.54 ^c^	0.54 ^d^	0.46 ^c^	0.62 ^c^	0.50 ^c^	n/a

Performance of models on a balanced subset of the test set (Dataset B, *n* = 668; 334 patients in each group). Performance was evaluated using the researcher nephrologists’ diagnosis as the ground truth. Values in the same column sharing a common superscript letter are not statistically significantly different (*p* ≥ 0.05). Pairwise comparisons were performed using McNemar’s test (Accuracy, Recall, Specificity), DeLong’s test (AUROC), and a bootstrap procedure (Precision, F1-score). Within the same column, values followed by the same superscript letter (e.g., a, b, ab) are not statistically significantly different (*p* ≥ 0.05). SVM, Support Vector Machine; LR, Logistic Regression; GB, Gradient Boosting; XGB, Extreme Gradient Boosting; RF, Random Forest; NB, Naive Bayes classifier; NN, Neural Network.

**Table 6 diagnostics-15-02801-t006:** Single machine learning models based on Dataset A (*n* = 1423 in each group) and Dataset B post-exclusion of patients without baseline creatinine values in the preceding 7 days as the testing dataset (*n* = 398).

Model	Accuracy	Precision	Recall	Specificity	F1 Score	AUROC
SVM	0.67 ^abc^	0.65 ^ab^	0.52 ^abc^	0.78 ^a^	0.58 ^ab^	0.71 ^b^
RF	0.67 ^ab^	0.65 ^ab^	0.54 ^ab^	0.78 ^a^	0.59 ^a^	0.71 ^ab^
GB	0.68 ^b^	0.67 ^bc^	0.54 ^ab^	0.80 ^a^	0.60 ^a^	0.71 ^ab^
NB	0.64 ^c^	0.73 ^c^	0.27 ^d^	0.92 ^b^	0.39 ^cd^	0.71 ^ab^
XGB	0.68 ^ab^	0.65 ^ab^	0.55 ^b^	0.77 ^a^	0.60 ^a^	0.70 ^ab^
LR	0.65 ^ac^	0.62 ^a^	0.49 ^ac^	0.77 ^a^	0.55 ^b^	0.69 ^a^
NN	0.61 ^e^	0.64 ^ab^	0.23 ^d^	0.90 ^b^	0.34 ^c^	0.64 ^d^
Traditional methods						
Computerized algorithm	0.94 ^d^	0.98 ^d^	0.88 ^e^	0.99 ^d^	0.93 ^e^	n/a
Clinician’s diagnosis	0.53 ^f^	0.46 ^e^	0.46 ^c^	0.59 ^c^	0.46 ^d^	n/a

Performance of models on a subset of the test set (Dataset B, *n* = 398) post-exclusion of patients without baseline creatinine values within the preceding 7 days. Performance was evaluated using the researcher nephrologists’ diagnosis as the ground truth. Values in the same column sharing a common superscript letter are not statistically significantly different (*p* ≥ 0.05). Pairwise comparisons were performed using McNemar’s test (Accuracy, Recall, Specificity), DeLong’s test (AUROC), and a bootstrap procedure (Precision, F1-score). Within the same column, values followed by the same superscript letter (e.g., a, b, ab) are not statistically significantly different (*p* ≥ 0.05). SVM, Support Vector Machine; LR, Logistic Regression; GB, Gradient Boosting; XGB, Extreme Gradient Boosting; RF, Random Forest; NB, Naive Bayes classifier; NN, Neural Network.

## Data Availability

The datasets used and/or analyzed during the current study is not freely available due to ethicak restrictions but may be available from the corresponding author Chung-Te Liu on reasonable request.

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
