# Peer review of "Machine Learning Models for Point-of-Care Diagnostics of Acute Kidney Injury"

_diagnostics, 2025, doi:10.3390/diagnostics15212801_

Round 1

Reviewer 1 Report

Comments and Suggestions for Authors

The paper aims to develop and evaluate machine learning models for diagnosing acute kidney injury (AKI) in hospitalized patients without available baseline serum creatinine (SCr). Using retrospective data from Wan Fang Hospital, the authors trained several ML models with point-of-care features and compared them to a computerized SCr-based algorithm and to clinicians’ diagnoses. The results show that repeated ML models achieved moderate accuracy, while single models varied. ML models outperformed clinicians but remained weaker than computerized SCr-based rules. The main contribution is the demonstration that ML can support AKI diagnosis when baseline SCr is missing, a situation common in practice.

The paper is interesting, but in its current form, it requires major revision before it can be accepted. My detailed comments are:

  1. The introduction mixes AKI prediction and diagnosis. The distinction should be explained more clearly.
  2. The study is based on a single center with relatively small test cohorts. Generalizability should be addressed and external validation is needed.
  3. The description of feature preprocessing, handling of missing values, and hyperparameter tuning is insufficient.

  4. Only accuracy, AUROC, and related metrics are given. Authors should also perform statistical comparisons between the models and the baseline algorithms.

  5. In the Discussion section, authors should explain how ML models will be used in real workflows and what added value they will bring beyond existing SCr-based alerts.

  6. Feature importance plots and results tables should be simplified and more clearly explained for readers without a technical background.

  7. More emphasis is needed on confounding by comorbidities and on the risk of misclassifying CKD as AKI.

Author Response

Comment 1: The introduction mixes AKI prediction and diagnosis. The distinction should be explained more clearly.

Response 1: Thank you for the comment. The text related to AI clinical prediction had been deleted (reference 16 and 17 in the prior version). Please verify that.

Comment 2: The study is based on a single center with relatively small test cohorts. Generalizability should be addressed and external validation is needed.

Response 2: Thank you for the comment. External validation is restricted by the institutional review board and is therefore not feasible. To compensate this, an independent testing dataset was used for validation (Dataset B). This had been mention in the paragraph of limitation on page 14, line 28 to 30.

Comment 3: The description of feature preprocessing, handling of missing values, and hyperparameter tuning is insufficient.

Response 3: We thank the reviewer for this important comment. We have added a detailed description of preprocessing, imputation, and tuning in the “Data Preprocessing and Imputation” subsection on page 7, line 17-19. Missing values were handled by mean imputation using training data only, and all features were standardized to zero mean and unit variance to prevent scale bias. Hyperparameter tuning—now described at the end of the “Development of machine learning models” section—was performed via randomized search with 5-fold cross-validation on the training dataset, and the optimal settings were used to retrain final models on the full training data. The above information had been added on page 7, line 33-38, and page8, line 8-26.

Comment 4: Only accuracy, AUROC, and related metrics are given. Authors should also perform statistical comparisons between the models and the baseline algorithms.

Response 4: We thank the reviewer for this valuable suggestion. We have revised the “Statistical analyses” section on page 9, line 23-30 to include formal statistical comparisons between all ML models and the two baselines (computerized algorithm and clinician diagnosis). Pairwise tests were conducted across six metrics under three test conditions using expert nephrologists’ diagnoses as ground truth. McNemar’s test was applied to accuracy, recall, and specificity; DeLong’s test to AUROC; and bootstrap analysis to precision and F1-score. Updated Tables 4–6 indicate significance using superscript letters. Results show all ML models significantly outperform clinicians (e.g., AUROC 0.74 vs. 0.53), while top models perform comparably. Relevant sections were updated to reflect these improvements.

Comment 5: In the Discussion section, authors should explain how ML models will be used in real workflows and what added value they will bring beyond existing SCr-based alerts.

Response 5: Thank you for the comment. The required text had been added in the second paragraph of the Discussion on page 12, line 15-18. (gold)

Comment 6: Feature importance plots and results tables should be simplified and more clearly explained for readers without a technical background.

Response 6: We thank the reviewer for this excellent suggestion. We agree that making our results more accessible to a broad clinical audience is essential. In response, we have substantially revised the captions for dimensionality reduction plot (Figure 1), the feature importance plot (Figure 2) and the results tables (Tables 4-6) to provide clear, non-technical explanations that will guide the reader in their interpretation. A more detailed interpretation for Figure 1 was also added to the results on page 10, line 13-20.

Comment 7: More emphasis is needed on confounding by comorbidities and on the risk of misclassifying CKD as AKI.

Response 7: Thank you for the comment. The present study aimed to diagnose AKI in patients with no known medical history who presented with abnormal SCr. Consequently, comorbidities were excluded from the feature set used by the machine learning model. This statement had been added in the section of Features of the machine learning models on page 6, line 14-16.

Reviewer 2 Report

Comments and Suggestions for Authors

Dear authors,

You wrote about the accuracy and the efficacy of ML to predict AKI.

  • The whole paper is complete and well-structured. For personal choice, I prefer to use the same dataset, splitting for train and test, and not two different datasets. Can you better detail this choice? 
  • You wrote "In each iteration, seven machine learning models were employed: the Support Vector Machine (SVM), Logistic Regression (LR), Gradient Boosting (GB), Extreme Gradient Boosting (XGBoost), Random Forest (RF), Naive Bayes classifier (NB), and Neural Network (NN)", without their details, such as kfolds, number of layer and leaf, number of ntree and mtree, ecc. You must add them.
  • Results are well reported

Summarily, the paper is well-structured, but it is not clear to clinicians. I think that you ought to change the line of the paper. Although it is correct and well structured, if you were to include a clinician in your audience of readers, you'd need to give the paper a different slant and include some ML info in the introduction. 

Author Response

Comment 1: The whole paper is complete and well-structured. For personal choice, I prefer to use the same dataset, splitting for train and test, and not two different datasets. Can you better detail this choice? 

Response 1: We thank the reviewer for this thoughtful comment. We intentionally used two temporally distinct datasets—Dataset A (2018–2020) for training and Dataset B (late 2020) for testing—to enable temporal validation, ensuring real-world applicability and robustness to data drift. This design better reflects clinical deployment, where models trained on historical data must perform on future patients. We have clarified this rationale in the “Study Design and Participants” section on page 5, line 15-20.

Comment 2: You wrote "In each iteration, seven machine learning models were employed: the Support Vector Machine (SVM), Logistic Regression (LR), Gradient Boosting (GB), Extreme Gradient Boosting (XGBoost), Random Forest (RF), Naive Bayes classifier (NB), and Neural Network (NN)", without their details, such as kfolds, number of layer and leaf, number of ntree and mtree, ecc. You must add them.

Response 2: We thank the reviewer for this valuable comment. We have revised the “Development of machine learning models” section to include full hyperparameter details on page 8, line 28-38 and page 9, line 1. Complex models were optimized through randomized search with cross-validation on the training data, then retrained on the full training set (Dataset A) and tested on Dataset B. We now report specific configurations for Random Forest, Gradient Boosting, XGBoost, and the Neural Network. For SVM, Logistic Regression, and Naive Bayes, we retained scikit-learn default parameters, as further tuning did not improve performance.

Comment 3: Results are well reported

Response 3: Thank you for the comment.

Comment 4: Summarily, the paper is well-structured, but it is not clear to clinicians. I think that you ought to change the line of the paper. Although it is correct and well structured, if you were to include a clinician in your audience of readers, you'd need to give the paper a different slant and include some ML info in the introduction. 

Response 4: Thank you for the comment. In response, text describing the scenario how our machine learning model may help inexperienced clinicians to differentiate AKI and CKD in case the patient did not provide any medical history had been added to the last paragraph of introduction on page 4, line 26-31.

Round 2

Reviewer 1 Report

Comments and Suggestions for Authors

The paper is not written in MDPI format. The authors addressed all my concerns. The paper can be accepted for publication.

Reviewer 2 Report

Comments and Suggestions for Authors

All suggestions have been solved